


# Vulnerability and Site Effects in Earthquake Disasters in Armenia (Colombia). II – Observed Damages and Vulnerability

by

Francisco J. Chávez-García[1], Hugo Monsalve-Jaramillo[2], Joaquín Vila-Ortega[3]

[1]Professor Instituto de Ingeniería, Universidad Nacional Autónoma de México, Ciudad Universitaria, Coyoacán 04510 CDMX, México paco@pumas.iingen.unam.mx

[2] *Correspondence to*: Professor Facultad de Ingeniería, Universidad del Quindío, Cra. 15 #12N, Armenia, Quindío, Colombia hugom@uniquindio.edu.co

[3]Professor Facultad de Ingeniería, Universidad del Quindío, Cra. 15 #12N, Armenia, Quindío, Colombia jjvilaortega@uniquindio.edu.co



**Abstract**

Damage in Armenia, Colombia, for the 1999 (Mw6.2) event was disproportionate. We analyse the damage report as a function of number of storeys and construction age. We recovered two vulnerability evaluations made in Armenia in 1993 and in 2004. We compare the results of the 1993 evaluation with damages observed in 1999 and show that the vulnerability evaluation made in 1993 could have predicted the relative frequency of damage observed in 1999. Our results show that vulnerability of the building stock was the major factor behind damage observed in 1999. Moreover, it showed no significant reduction between 1999 and 2004.

Key words: earthquake damage; vulnerability; construction type; construction age; building inventory.

# 1 Introduction

Destructive earthquakes occur relatively frequently in Colombia (the first reported event dates from 1551, Espinosa, 2003). However, the development of earthquake engineering began only relatively recently, punctuated by several major, significant events. The first building code in the country was published in 1984 (CCCSR-84, 1984), partly as a result of the heavy toll caused by the Popayán earthquake in March, 1983 (Ingeominas, 1986). Increasing building requirements have improved earthquake resistance, for example phasing out non engineered construction. The development of earthquake engineering has led to a decrease in the vulnerability of buildings in Colombia but progress has been slow, in pace with the development of building codes. In addition, as favoured construction styles evolve, additional challenges appear. For example, the cost of land pushes current housing projects consisting of tall concrete structures for which there is little experience regarding their seismic behaviour in that country. Instrumenting some of those buildings to analyze their motion during small earthquakes would provide useful data and may eventually become a necessity (e.g., Meli et al., 1998). Meanwhile, it is important to learn as much as possible from past destructive events.

Damage evaluation after large earthquakes is recognized as a primary input to understand structural response subject to dynamic excitations. It offers valuable data on the behaviour of structures to actual seismic motion. In addition to very significant efforts like GEER (Geotechnical Extreme Events Reconnaissance, 2020), local initiatives have contributed significantly to understand damage occurrence, especially in relation to site effects (e.g., Montalva et al., 2016; Fernández et al., 2019).

One seismic event that has had a long lasting impact in Colombia is the January 25, 1999, earthquake in the Quindío department, close (18 km) to the city of Armenia. This relatively small (Mw6.2), normal fault earthquake had profound economic and social consequences in the country. There was only one accelerograph in Armenia, and it recorded PGA of 518/580/448 gal in the EW/NS/Z components. Strong ground motion duration was very short (smaller than 5 s) and ground motion energy peaked at periods shorter than 0.5 s. The source of the main shock and aftershocks was studied in Monsalve-Jaramillo and Vargas-Jiménez (2002), while macroseismic observations were presented in Cardona (1999). The city of Armenia sustained heavy damage (maximum intensity was IX in EMS-96 scale): 2000 casualties and 10,000 injuries due to the collapse of 15,000 houses, with a further 20,000 houses severely damaged (SIQ, 2002). Site effect evaluation during this event in Armenia was addressed by Chávez-García et al. (2018). Earthquake and ambient noise data were analysed with the objective of characterizing local amplification due to soft surficial layers using a variety of techniques. The results showed that, while local amplification contributed significantly to destructive ground motion, observed damage distribution in 1999 was incompatible with the rather small variations in dominant frequency and maximum amplification throughout the city.

Chávez-García et al. (2018) referred to the damage distribution observed for the 1999 earthquake but no data were analysed in that paper. In this paper, we present an analysis of damage observed during the 1999 earthquake. Earthquake damage data is analysed in relation to geology and to the site classes defined in the microzonation map of Armenia (Asociación Colombiana de Ingeniería Sísmica, 1999). In addition, the city of Armenia offers a very uncommon advantage in Latin America. Two vulnerability studies have been conducted in the city, one in 1993 and one in 2004. We compare the 1999 damage distribution to vulnerability estimated in 1993 for the small downtown district of the city where the two data sets overlap. The comparison of the two vulnerability studies, in 1993 and in 2004, allows an assessment of the changes in vulnerability in the city as a consequence of a destructive earthquake, even if the method used was different and the studied zones overlap only partially. We show that building vulnerability was the main factor behind the heavy damage toll in Armenia during the 1999 earthquake. Our results substantiate the improvement of engineering practice with time and provide evidence of the efficacy of simple methods to evaluate vulnerability. However, they also strike an alarm bell as they show that vulnerability in Armenia remains high. Our results offer an unusually complete analysis of the major factors behind seismic risk in a typical medium size city in Colombia. Seismic risk mitigation in Armenia, and in similar midsize cities in Latin America, requires an increase in the number of permanent seismic stations and support of additional efforts to improve our understanding of moderate size seismic events.

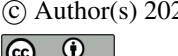


## 2 Colombian Building Codes and Practice Evolution

This paper will obviate a discussion of the geological setting of Armenia, as it can be found in Chávez-García et al. (2018).
The coffee growing region was occupied during the second half of the 19th century. For this reason, data on historical
earthquakes is scarce, even though it is located in a zone of high seismic hazard (the current Colombian building code
prescribes a PGA of 0.25 g for Armenia for a return period of 475 yr). During the 20th century, about eight earthquakes
occurred in the region producing intensities as large as IX (Espinosa, 2011).
Before 1960, construction in this region consisted mainly of bahareque and unreinforced masonry. In Colombia,
bahareque refers to structures that use guadua (a local variety of bamboo) for the skeleton elements. Walls are made using
a guadua-based mat, covered with mud mixed with dung as bonding agent. At about 1960, reinforced concrete frames
began to be used but Colombia lacked a building code until 1984, although conscientious engineers followed guidelines
from international codes, mostly American ones. Between 1977 and 1984 design practice for those structures shifted from
the elastic method to ultimate strength design. Unfortunately, this allowed construction companies to decrease the quantity
of steel reinforcement. Until 1984, no seismic provisions were considered.
A major milestone was the Popayán earthquake of March 31, 1983 (ML5.5). This small, shallow event caused major
destruction in Popayán, where important Spanish heritage sites were severely damaged. Although restricted in extension,
the heavy damage gave the final push for the adoption of a national building code including seismic provisions in 1984.
This code had been promoted since the end of 1970's by Asociación Colombiana de Ingeniería Sísmica (Colombian
Association for Earthquake Engineering), founded on 1975. A major consequence of the 1984 code was to eliminate new
construction using unreinforced masonry. This code was replaced by a new version in 1998. The effects of two events in
1995 (Feb 8, Mw6.6, Aug 19, Mw6.5) convinced engineers that lateral drift requirements in the 1984 code were too
lenient and stricter requirements were incorporated.
Only a few months passed between publication of the 1998 building code and the occurrence of the 1999, Armenia,
earthquake. Some of the shortcomings identified during this event were addressed in improvements to the code published
in 2010; requisites for irregular buildings with weak storeys, short columns, p-Δ effects, and torsion related problems
among others. Microzonation of cities with more than 100,000 people became mandatory. However, those studies are the
responsibility of local authorities and are not necessarily considered a priority. In Armenia, nine years after becoming
compulsory, an update of the microzonation study carried out in the wake of the 1999 earthquake is still missing.
Currently, discussions for a new version of the building code centre on imposing requisites on the quality control of the
materials used and ensuring the correspondence between drawings and the real structure.

## 3 Damage Observed in 1999

In the aftermath of the 1999 event, the Sociedad de Ingenieros del Quindío (Quindian Society of Engineers) organised
teams that made a detailed evaluation of damaged structures in Armenia (SIQ, 2002). The status of a building is
determined by the attributes of damage level, damage type and usage status (Tang, et al., 2020). The priority was to
distinguish between those buildings that did not pose a risk to occupants from those that must be evicted. The template
used to qualify buildings allowed to grade the damage sustained by buildings and included information on year of
construction, structural system, and number of storeys. SIQ (2002) classified observed damage using a colour scale:
●   Grey. Very light or no damage at all.
●   Green. The building can continue to be used. Although some damage is apparent in non-structural elements, it
160         poses no risk to occupants
●   Yellow. Significant damage, to the point that partial occupancy restriction is required. The structure is not
162         evaluated as unsafe but access to parts of it must be restricted.
●   Orange. Unusable structure. Damage to the structure implies a high risk and the building cannot be occupied.
●   Red. Total collapse or danger of collapse due to severe damage to the structure or its foundation.
This scale is quite standard and very similar to that proposed by the European Seismological Commission (Xin et al.,
2020). For our purpose, we have simplified this scale. We use light damage to refer to structures classified in grey or
green. Moderate damage in this paper is used for buildings classified as yellow. Finally, severe damage corresponds to
structures classified as orange or red. The SIQ (2002) report presents an inventory of 43,023 structures classified as a
function of damage sustained. From this total, data for 1,946 sites could not be used due to incomplete information that
made it impossible to locate them on a map. This number suggests a lower limit for the uncertainties in our database,
inevitable in any post-earthquake damage survey and which we have no means to evaluate. However, the number of
samples is large enough to justify our confidence in average values. Our final database for Armenia includes 41,077
buildings. Data is available only for damaged structures and it is not possible to normalize the results relative to the
number of existing buildings in the city.
Five categories were used to classify the buildings structuring type, following CCCSR-84 (1984). In order of decreasing
seismic performance, the first four categories are: frame structures, confined masonry, unreinforced masonry, and
bahareque structures (wooden structures are included here). The fifth category, as written in the template used by SIQ



(2002), is "none of the preceding", named as "other" in the following. This last category was used to refer to buildings
using hybrid structuring systems, a mix of different materials, and unstructured houses mixing wood with other elements.
Such precarious houses are non-engineered structures and are common in illegal settlements.
Figure 1 shows the distribution of the 6,467 structures classified as severely damaged in Armenia. The background of the
figure shows the geological formations that can be found in the city (AIS, 1999). No correlation is observed between
geology and severe damage distribution. The same observation can be made for moderate and light damage. Site geology
seems irrelevant to explain damage distribution for this event, which shows no clear pattern. It may be argued that the
geological classification cannot reflect site effects caused by mostly thin layers. That site effects in Armenia are related
to thin layers is suggested by the values of dominant frequencies in the city, shown to be comprised between 2 and 3 Hz
by Chávez-García et al. (2018). Figure 2 shows the depth to the base of ash deposits in Armenia (Ingeominas, 1999),
determined from the inversion of 36 vertical electrical soundings. Dominant frequencies computed for the thicknesses
shown in Figure 2 using the average shear-wave velocities for the topmost sedimentary layers (Chávez-García et al.,
2018) are comprised between 2 and 3 Hz, similar to those observed. Shallow soils in Armenia were mapped in the
microzonation study of the city, carried out in the wake of the 1999 earthquake (AIS, 1999). The final microzonation map
proposed by AIS (1999) proposed four different soil types: ash deposits (zone A), thin sedimentary fill deposits (zone B),
alluvial terraces, residual soils and ancient volcanic flows (zone C), and soils that have undergone shearing as they are
located close to Armenia fault that cuts through the city (zone D). The seismic coefficients proposed by AIS (1999)
decrease from zones A to C, implying similarly decreasing site amplification. Zone D was declared inapt for construction.
Figure 3 shows histograms of damage distribution for the city as a function of structuring type, damage level, and soil
class. Bahareque structures suffered the largest proportion of severe damage, followed by structures in the category
"other" and unreinforced masonry. Figure 3 shows clearly that damage distribution is independent of soil type as classified
by the seismic microzonation study. This result supports the conclusions of Chávez-García et al. (2018). They observed
that, while local amplification is far from being negligible, it does not vary greatly within Armenia.
Consider now the role of two additional variables on damage distribution: number of storeys and building age. In order
to compare these results with the vulnerability study made in 1993 in Armenia, we restrict this analysis to the small
downtown district shown in Figure 2, where the 1993 study was carried out. In this sector, the damage database includes
3,697 records corresponding to 470 bahareque, 884 unreinforced masonry, 195 confined masonry, and 745 frame
structures. We dropped the data for 1,403 structures classified as "other". Figure 4 shows damage distribution as a function
of number of storeys and structuring type. The diagram for all types of structures combined shows an apparent decrease
in severe damage and increase in light damage with increasing number of storeys. The diagrams for each structure type
do not show such progression. The reason for that apparent trend is that buildings smaller than five storeys are
overrepresented (90% of our sample) in the downtown district. One- to two-storey high bahareque structures are 95% of
the total. The tallest unreinforced masonry structures were one 6-storey and one 10-storey buildings. With this caveat, it
is clear that number of storeys was not a major factor in damage distribution during the 1999 earthquake in Armenia.
Figure 5 shows damage distribution as a function of structuring type and construction period, again for the small
downtown district. Our division of time corresponds to the evolution of construction practice in Colombia, as discussed
above. Severe damage in bahareque structures do not show a clear trend with time; it is larger than 60% for all periods,
except for the period 1985-1997. The period later than 1998 is not representative for bahareque structures as there is only
one light, zero moderate, and two severely damaged structures. In contrast, severe damage for frame and unreinforced
masonry structures shows a steady decrease with time (and the number of structures is significant). The relative number
of structures suffering light damage increases with decreasing age of the structure, while the relative frequency of severe
damage decreases significantly, showing the benefit of building code improvements. The number of confined masonry
structures built before 1959 was very small (10 buildings in our sample) making the histograms for that period unreliable.
For later periods, confined masonry shows an increase in the percentage of light damage and a stable or decreasing
percentage for moderate and severe damage.
**4 Vulnerability and Damage Distribution**
Earthquake damage is the result of strong ground motion and building vulnerability. Vulnerability of the building stock
has always been a key factor in seismic risk evaluations (e.g., Dolce et al., 2006; Vicente et al., 2014; Fikri et al., 2019),
or post-earthquake evaluations (e.g., Marotta et al., 2017). A review of current challenges has been presented in Silva et
al. (2019). A major problem is the large number of buildings for which a vulnerability estimate is required in a city. When
the number of structures is limited to a few hundreds, simple methods are often used, which usually consist in simple
evaluations of a limited number of parameters (e.g., Fikri et al., 2019). Larger building populations have to be dealt with
using probabilistic methods (e.g., Noh et al., 2017) or extremely indirect techniques (Geiß et al., 2014).
In Latin America, vulnerability studies of the building stock are not often made outside capital cities. However, in the
case of Armenia, we are fortunate to have available two vulnerability studies: one performed in 1993 (López et al., 1993),
six years prior to the 1999 event, and one made in 2004 (Cano-Saldaña et al., 2005). Those two studies followed different
procedures and the area coverage overlaps only partially (Figure 2). In this section, we will compare the results of the


1993 vulnerability study with damage distribution observed in 1999. Then, we will compare the two vulnerability
evaluations between them.
In 1993, different sectors of the city were sampled but not all of the data were preserved. We analyse the results for the
downtown sector presented in López et al. (1993), shown in Figure 2. A census was made to count the number of structures
of each type. In the downtown sector, 3,364 buildings were counted and assigned to one of three categories: bahareque
structures (908), unreinforced masonry structures (1,877), and frame structures (579). It was not possible to evaluate,
even in a simplified way, all those structures. For this reason, a small sample of 84 buildings was designed, assuming
normal distribution and choosing a 95% confidence level of the extrapolation of the results to the total population. The
84 buildings were randomly selected in the field and the vulnerability of each of them was evaluated using the procedure
described in Tassios (1989), which is very similar to that described in Inel et al. (2008) or Alam et al. (2013). Each selected
building was visited by a team of students of civil engineering and a detailed template was completed with information
on the structure. The compiled information consisted of: structuring type, relation with neighbouring structures (possible
interaction problems), year of construction, maintenance, vertical and horizontal configuration, and roofing material.
These factors were assigned numerical values and combined with arbitrary weights based on expert opinions to compute
a vulnerability index (VI) for each building. VI was made to vary between 0 and 100, where 0 corresponds to an absolutely
safe structure and 100 to a totally vulnerable structure. Finally, the vulnerability indexes determined for the sample were
extrapolated to the complete population in the downtown district.
Figure 6 compares the VI values determined in 1993 with damage observed during the 1999 earthquake inside the
downtown district (solid line polygon in Figure 2). Percentages for VI values were extrapolated from the numbers
determined for the 84 building sample. In this figure, we counted together moderate and severe damage, while VI was
classified in two groups: larger and smaller than 20. We observe a very good correlation between VI estimated in 1993
and damage observed during the 1999 earthquake, six years later. Thus, the approximate procedure used to estimate VI
in 1993 was effective to predict dynamic behaviour during that earthquake.
In addition to comparing extrapolated VI with damages for the downtown district, we may ask another question. How did
each one of the 84 buildings, whose VI was evaluated, fare during the 1999 earthquake? This question has no simple
answer due to different georeferencing systems for the two surveys (vulnerability and damage) and incomplete data. Only
28 out of the 84 could be confidently identified. The unidentified buildings could be absent from the damaged buildings
database because they suffered no damage or because their recorded location was inaccurate. Figure 7 shows a whisker
plot of the observed VI values against observed damage for the 28 buildings that could be identified in both databases.
VI values are well correlated with observed damage. Figure 7 shows that severe damage may be associated with an
average VI of 44, moderate damage with an average VI of 32, while light damage corresponds to an average VI of 16.
Consider finally the vulnerability study made in 2004 (Cano-Saldaña et al., 2005). The procedure used was very different
and followed that of Velásquez and Jaramillo (1993). Cano-Saldaña et al. (2005) computed expected losses for three
different events, considered to pose the largest seismic hazard for Armenia. A required input for them was an estimate of
the vulnerability for the building stock, and this is the data we recuperated from that study. Cano-Saldaña et al. (2005)
selected a sector of the downtown district that overlaps only partially with the district sampled in 1993. It is shown with
dot-dashed line in Figure 2. They tallied every building in that sector, a total of 2,525 land plots. For each one of them, a
template simpler than that of 1993 was completed including data on structuring type, number of storeys, roofing type,
and construction quality. The simplified nature of the template made it possible to complete it for the 2,525 land plots, in
contrast to the more detailed template used in 1993. We recuperated the 2004 building database and estimated
vulnerability using the same procedure used in 1993; i.e., assigning numerical values to each factor and combining them
with arbitrary weights based on expert opinions to compute a vulnerability index for each building in the sample. The
weights used to estimate a vulnerability index had to be modified from those used in 1993 given that less information on
each structure was available. The VI results for the 2004 study may thus have a constant bias. We could assign a
vulnerability index to 1,217 buildings, out of the 2,525 counted in 2004. The building categories that could be identify
between the two studies were bahareque, unconfined masonry and frame structures. VI values were grouped in three
categories: low (VI between 0 and 20), medium (VI between 20 and 40), and high (VI larger than 40). The results in
Figure 8 show that the relative proportions are maintained between 1993 and 2004: most buildings in that sector have still
high vulnerability in 2004 and less than 20% have low VI. Our results suggest that significant improvements in the relative
vulnerability occurred in the 11-year period between 1993 and 2004. High vulnerabilities are still predominant in
downtown Armenia, in spite of the destruction of weak buildings in the 1999 earthquake and the reinforcement carried
out during the reconstruction of the city. It may be hoped that this result will prompt local authorities to take decisive
actions to mitigate seismic risk in Armenia. A starting point could be to replicate the use of simplified procedures to
estimate vulnerability to evaluate possible changes in the 16-year period since 2004.
**5 Conclusions**
Colombia, and in particular the coffee growing region, has been historically affected by large earthquakes, with the 1999
event being the most recent destructive event. The consequences of that earthquake significantly changed society in





Armenia and forced important improvements in engineering practice. The large economic consequences led the
government to add a new tax to pay for reconstruction: a levy of 2‰ was imposed on every bank transaction in the
country. Earthquake disasters occur rarely and therefore seismic risk is seldom a priority. In Armenia region, the first two
accelerographs were installed in 1994: in the campus of Universidad del Quindío, and in Calarcá (a neighbouring town,
10 km to the SE of Armenia). To date, they continue to be the only accelerographs in operation. As mentioned above, the
mandatory microzonation study of Armenia is still due.
We have presented an analysis of observed damage and vulnerability in Armenia during the 1999 earthquake. Our results
are based on databases that had remained as unpublished reports. The severity of damage is uncorrelated either with
geology or with the zones identified in the microzonation map. Damage distribution is uncorrelated with structure height
but we do observe a decrease in the severity of damage for younger structures. The data on observed damages were
contrasted against two vulnerability evaluations, one in 1993 and one in 2004. In the 1993 study, 84 buildings were visited
and their vulnerability was evaluated using a detailed template. The comparison of the results with observed damage in
the city six year later strongly supports this method.
Our results indicate that building vulnerability was the main factor behind the large damage caused by the 1999
earthquake. The comparison between the vulnerability studies of 1993 and 2004 show no significant improvements in the
relative vulnerability in that 11-year period. Unfortunately, it is possible that the money allocated to house owners for
repairs may not have been used to that purpose. Seismic risk mitigation in Armenia, and in similar midsize cities in Latin
America, requires more decisive support to increase the number of permanent seismic stations. This is especially
important given that current practice fosters tall concrete structures for which there is little experience regarding their
seismic behavior. This paper strives to ring an alarm bell to the current risk in Armenia through a better understanding of
a significant past destructive event.

**Author Contributions:** HMJ recovered the original data. FJCG, HMJ and JJVO analyzed the data. JJVO prepared the
maps and processed the statistics of the data. FJCG wrote the first draft and prepared the figures. All three authors revised
the manuscript and made final corrections.
**Competing interests.** The authors declare that they have no conflict of interest.
**Acknowledgements**
We thank Sociedad de Ingenieros del Quindío and Servicio Geológico Colombiano (ancient Ingeominas) for authorization
to use the data included in the unpublished reports on damage distribution and microzonation projects carried out in
Armenia after the 1999 earthquake. Part of this research was made during a sabbatical visit of FJCG to Facultad de
Ingeniería, Universidad del Quindío. This visit was possible thanks to the support of Dirección General de Asuntos del
Personal Académico of UNAM through program PASPA. Additional support was received from project 934 supported
by Vicerrectoría de Investigaciones of Universidad del Quindío, and from the Rectory of Universidad del Quindío.

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

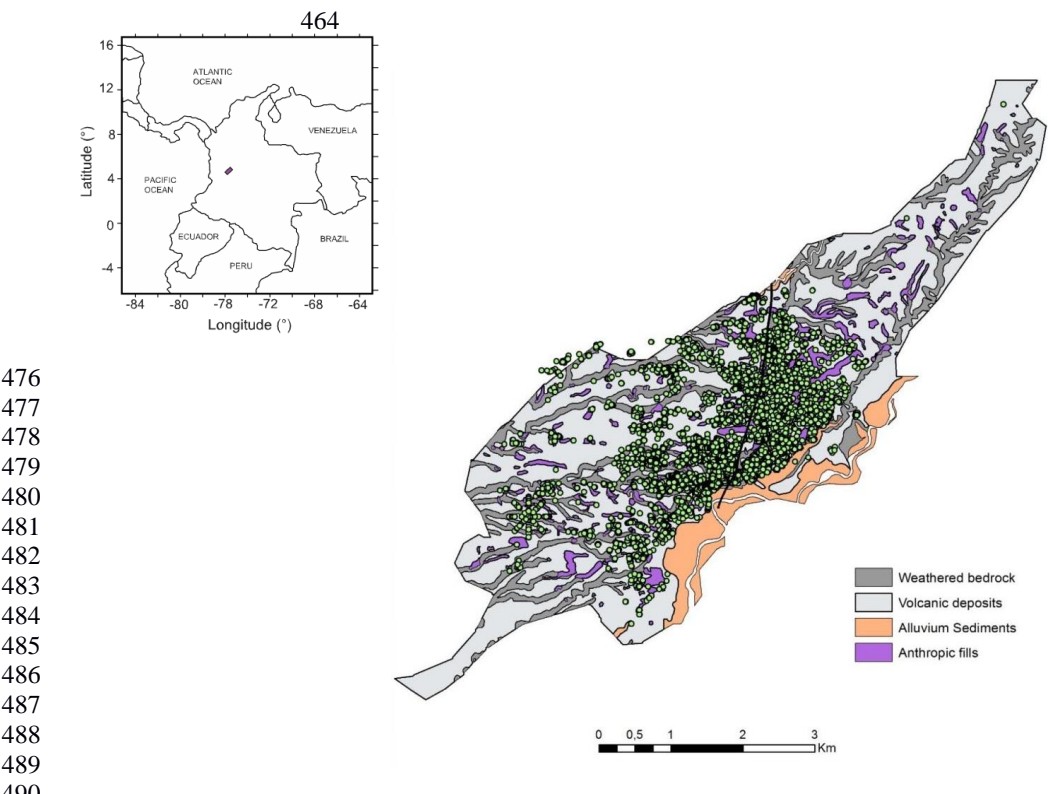

Figure 1 Upper left: The small rectangle shows the location of Armenia in Colombia, South America. The main figure
shows the geological map of the city. The small circles indicate the location of 6,467 structures that were severely
damaged during the January 25, 1999, earthquake. The thick solid line crossing the city from north to south shows the
trace of Armenia fault. [Modified from Ingeominas, 1999.]

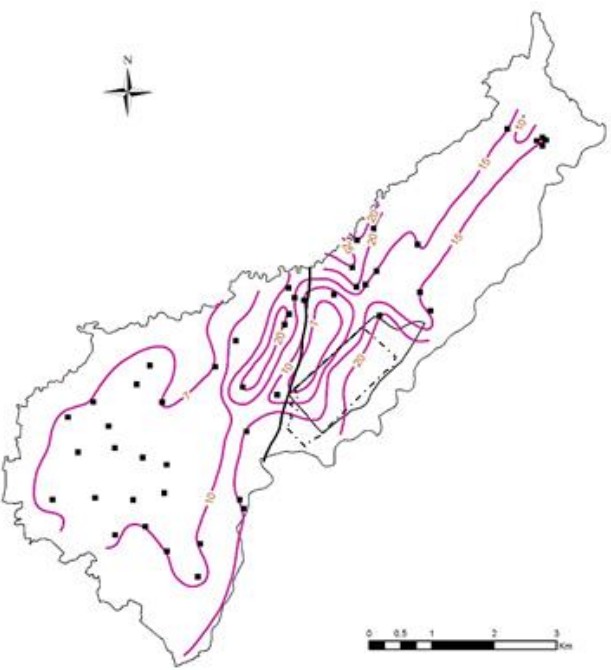

Figura 2 Contornos de la profundidad de la interfaz (en m) en la base de los depósitos de ceniza que cubren la ciudad de Armenia. Los
símbolos muestran la ubicación de los 36 sondeos verticales eléctricos donde se midió la profundidad de esa interfaz. La gruesa línea
sólida que cruza la ciudad de norte a sur indica el rastro de la falla de Armenia. El polígono de línea sólida dentro de la ciudad muestra
la extensión del distrito del centro cubierto en el estudio de vulnerabilidad de 1993. El esquema de línea pequeña y salpicada de puntos
muestra el área cubierta por el estudio de vulnerabilidad realizado en 2004. [Modificado de Ingeominas, 1999.]




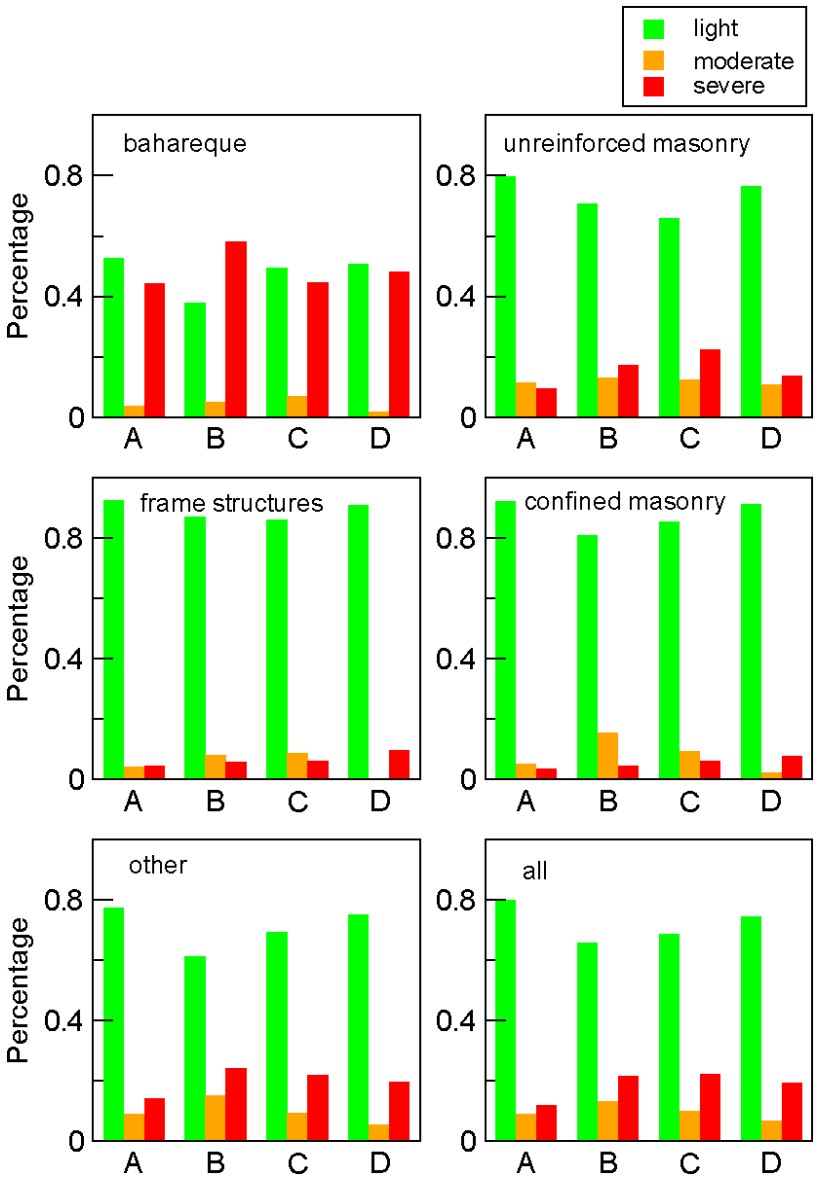

Figure 3 Histograms of observed damage in Armenia for the January 25, 1999, earthquake. Each diagram corresponds to
the given structuring type and shows the relative incidence of light, moderate, and severe damage as a function of the four
soil types defined in the microzonation map of AIS (1999) (A, B, C, and D). The last diagram shows data for all structuring
types together.




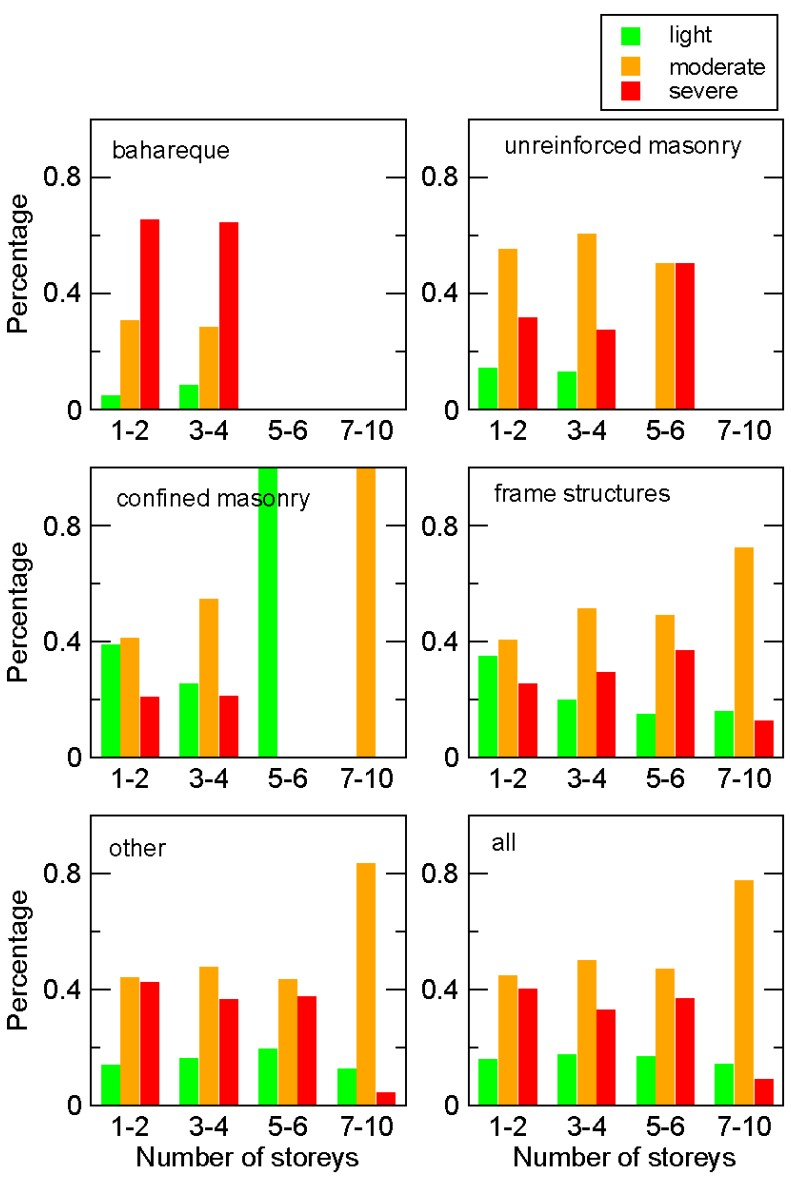

Figure 4 Histograms of observed damage in Armenia for the January 25, 1999, earthquake. Each diagram corresponds to
the given structuring type and shows the relative incidence of light, moderate, and severe damage for groups of buildings
of similar number of storeys. The last diagram shows data for all structuring types together.



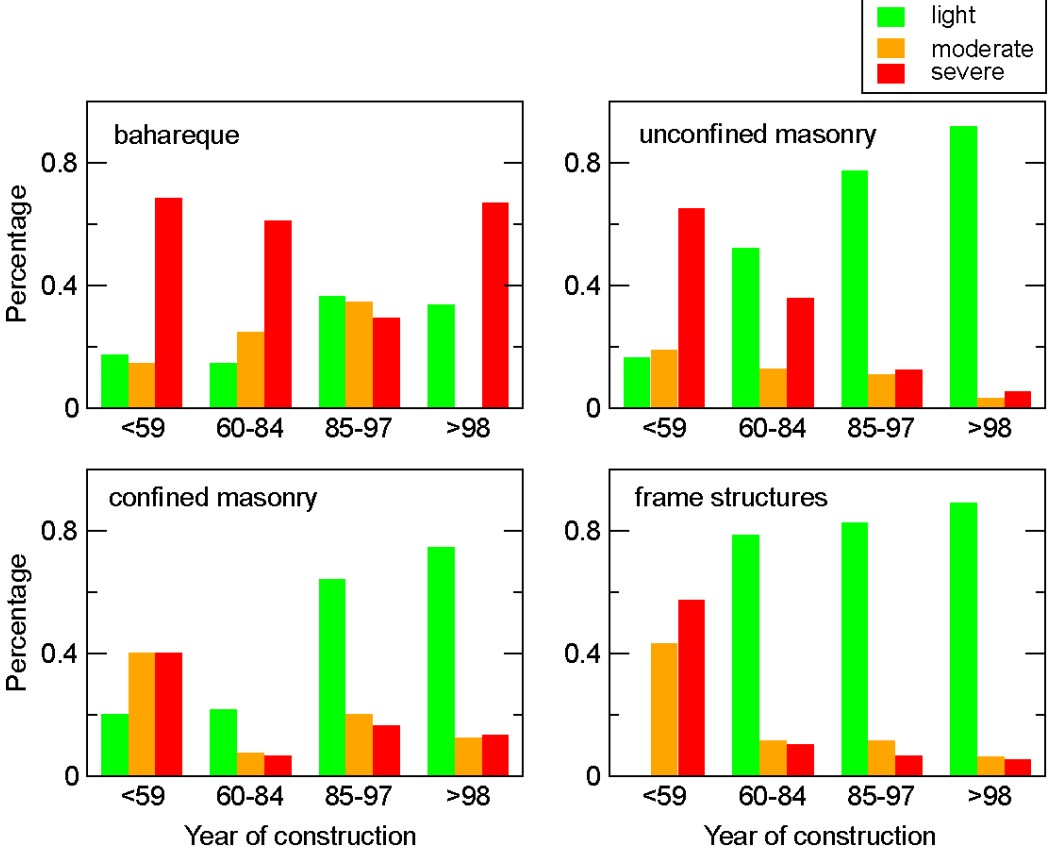

Figure 5 Histograms of observed damage in Armenia for the January 25, 1999, earthquake. Each diagram corresponds to
the given structuring type and shows the relative incidence of light, moderate, and severe damage as a function of the
time period where the structure was built (before 1959, between 1960 and 1984, between 1985 and 1997, and later than
1998). The data shown corresponds to the downtown district whose outline is shown in Figure 2.


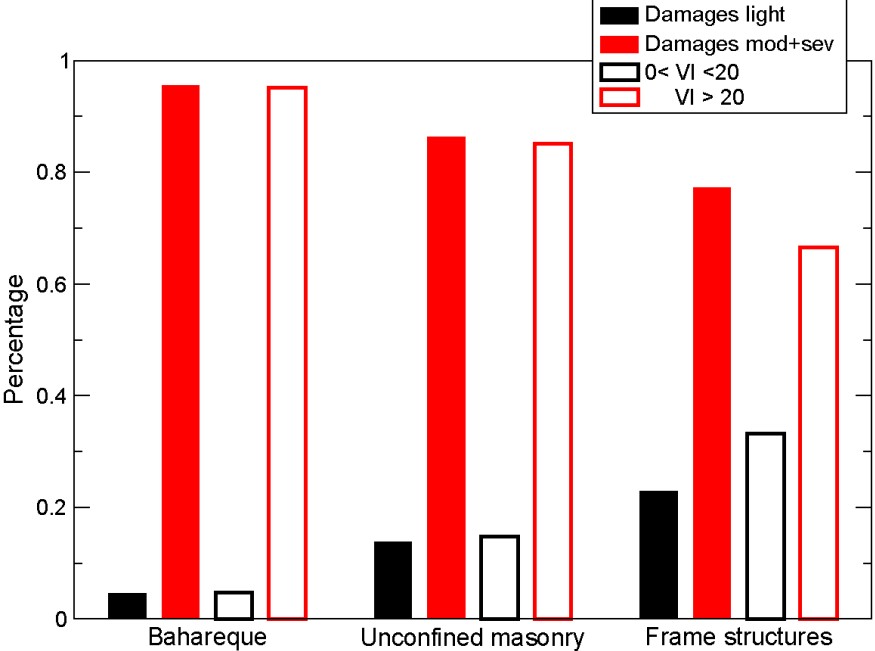

Figure 6 Comparison between vulnerability values estimated in 1993 and damages observed in 1999. This comparison
was only possible for the three structuring types shown. Moderate and severe damages were counted together.
Vulnerability indexes (VI) are separated in two groups, below and above a value of 20. Both damages and vulnerabilities
correspond to the complete building population inside the polygon drawn with solid line in Figure 2.




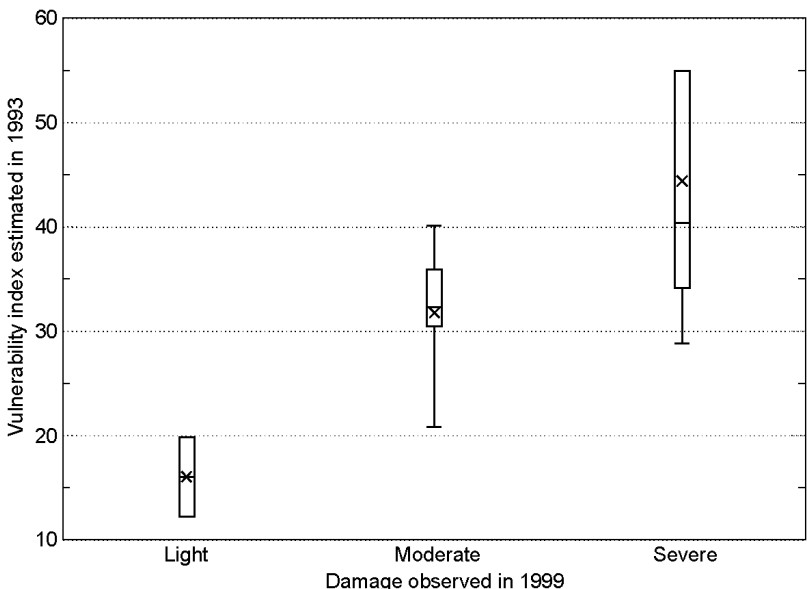

Figure 7 Whisker plot comparing the vulnerability index for 28 buildings evaluated in 1993 against their actual behavior observed during the 1999 earthquake. The cross inside each symbol indicates the location of average values.




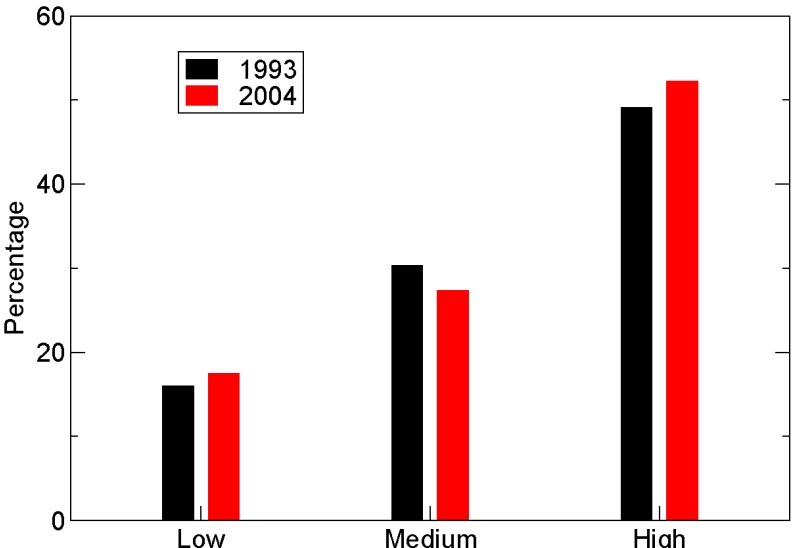

Figure 8 Comparison between percentages of buildings classified as low, medium and high vulnerability between the
evaluation made in 1993 and that of 2004 in Armenia. The values for 2004 used ad-hoc weights in an effort to get a
vulnerability estimate compatible with the scale used in 1993. Values for 2004 may thus have a constant bias.