# Peer review of "Vulnerability and Site Effects in Earthquake Disasters in Armenia (Colombia). II – Observed Damages and Vulnerability"

_Natural Hazards and Earth System Sciences, 2020_

## Referee Comment (RC1) · Anonymous Referee #1 · 1 Mar 2021

The paper entitled "Vulnerability and Site Effects in Earthquake Disasters in Armenia (Colombia). II - Observed Damages and Vulnerability" is a research based on analysis of vulnerability e site effects in the area of Armenia (Colombia). The manuscript represent a substantial contribution to the understanding of natural hazards and their consequences. The scientific and technical approaches and the applied methods is valid and presentation of scientific data, results and conclusions needs to be improved.

- The paper can be published after some corrections. - Abstract: Enter the dates and main characteristics of the earthquakes indicated. - Page 3: Insert a table of the earthquakes described. - Page 4: Insert some stratigraphic columns and shear wave

velocity profiles. - Figure 2: Enter the text in English.

Please also note the supplement to this comment:
https://nhess.copernicus.org/preprints/nhess-2020-385/nhess-2020-385-RC1-
supplement.pdf

---

## Author Comment (AC1) · 16 Mar 2021

Answer to the comments by Anonymous Referee #1

First we would like to thank Anonymous Referee #1 for reviewing our manuscript. We thank him/her for his/her positive appreciation of our manuscript. His/her review raises a few points where our manuscript needs to be improved. We recall those points and indicate the way in which we are modifying our manuscript to solve the Referee's concerns.

Abstract: Enter the dates and main characteristics of the earthquakes indicated.

NHESS imposes limits to the Abstract length. However, we have added some additional details of the earthquake that is mentioned there.

Page 3: Insert a table of the earthquakes described. A table is being added to the manuscript with the requested data.

Page 4: Insert some stratigraphic columns and shear wave velocity profiles. We have prepared a table with data of the soil profile at two sites in the city of Armenia. Those two sites are representative of the range of variations of shear wave velocity and soil thickness throughout the city. This addition requires the modification of Figure 2, where the location of those two sites will be indicated.

Figure 2: Enter the text in English. We do apologize for this oversight. We do not understand how the caption for Figure 2 managed to slip into the submitted document in Spanish. This blunder has been corrected.

We thank again Anonymous Referee #1 for his remarks that are helping us to improve our manuscript. A revised version will be uploaded after the review process is completed.

---

## Referee Comment (RC2) · Francesco Panzera (Referee) · 30 May 2021

The manuscript "Vulnerability and Site Effects in Earthquake Disasters in Armenia (Colombia). II – Observed Damages and Vulnerability" by Chávez-García et al. summarize the damage studies performed in Armenia (Colombia) trying to explain the main factors which played a role in the damage distribution. The manuscript can be accepted after minor revisions.

Francesco Panzera

Comments: Line 57: add "of buildings" after "number of storeys and construction age"
Line 82: add some worldwide references. I suggest: Midorikawa (2002) Ann. Geophys. 45 (6), 769–778 Sbarra et al (2012) https://doi.org/10.1007/s11069-012-0085-9 Panzera et al. (2018) https://doi.org/10.1016/j.enggeo.2018.04.014 Panzera et al. (2021) https://doi.org/10.1007/s10950-020-09962-z; Line 86: change "relatively small" with "moderate earthquake" or "strong earthquake". Line 184: "the geological formations that can be found in the city", please specify the used geologic map scale. General comment: The authors excluded completely the influence of site effects, but in my opinion some sentences on the fundamental periods of buildings in Armenia should be discussed to completely exclude resonance effects or to verify that the combination of vulnerability of buildings and site effects enhanced the damage. The fundamental frequencies observed in the area (2-3 Hz) by Chávez-García et al. (2018) for site effects could interact with buildings having 2-5 storeys. This doesn't mean that the damage are mainly due to site effects, but it is possible that in some areas they contributed to damage (see the example of Carlentini and Lentini Panzera et al. (2018)).

---

## Author Comment (AC2) · 5 Jun 2021

Authors' answer to the comments to our manuscript by Dr. Francesco Panzera

First, we would like to thank Dr. Panzera for his review of our manuscript. His report includes a general evaluation of our manuscript, four comments pointing to details that could be improved in our text, and a general comment concerning a more significant topic. In the following, we recall each of the reviewer's objections and describe the way in which a possible revised manuscript will correct our manuscript taking into account the reviewer's comments. We have been instructed by the editor to delay the

preparation of a revised manuscript.

General evaluation of the manuscript. Dr. Panzera first describes briefly the contents and intent of our manuscript. We thank him for his positive judgment of our work.

Four minor comments. Nhess-2020-385-RC2 document makes four suggestions to improve our presentation. Three of them are wording corrections that will improve the clarity of the text. The fourth suggestion is to include additional references. The referee suggests four recent papers that cover similar subjects as our manuscript and highlight cases that occurred in areas removed from our case of study. We thank Dr. Panzera for providing the links to the publications, which allowed us to get those papers immediately. We agree that adding these references to our manuscript broadens the geography of referred papers and simplifies the way (through the references included in those four papers) for interested readers to access more papers on our subject.

General comment. Finally, in a general comment, Dr. Panzera suggests us to improve the discussion around the possibility of site effects contributing to irregular damage distribution. We thank him for this remark that identifies a point where our manuscript could improve. First, as our title suggests, this manuscript is the second part of a study on ground motion of the 1999, Armenia, earthquake (part I) and the understanding of the factors which played a role in the damage observed during that event. However, the general comment from referee number 2 indicates that our manuscript lacks clarity regarding the role of site effects. True, a more detailed account of that is presented in part I of our study. However, part II requires to be self-contained. In this sense, some additional comments regarding the possible double resonance (soft soil resonance coupled to building resonance) would certainly improve our manuscript.

Finally, we thank again Dr. Panzera for his remarks that are helpful to improve our manuscript.

---

## Author Response (AR1)

REPLY TO REVIEWS

This document presents a point-by-point reply to the comments received by our manuscript during the review process. We benefited from detailed reviews by two reviewers, one anonymous, one signed. In the following, we recall the comments to our manuscript in each review and describe the way in which each one of them was considered to prepare the revised version of our manuscript. We took the opportunity of this revision to check again our text. We corrected typos and improved English usage in three or four places through different word choices.

**Comments by Anonymous Referee #1**

First we would like to thank Anonymous Referee #1 for reviewing our manuscript. We thank him/her for his/her positive appreciation of our manuscript. His/her review raised a few points where our manuscript needed to be improved. We recall those points and indicate the way in which we modified our manuscript to solve the Referee's concerns. The changes made to the manuscript to solve this referee's concerns have been highlighted in yellow in the marked-up version of our revised manuscript.

*Abstract: Enter the dates and main characteristics of the earthquakes indicated.*
We have added additional details of the earthquake that is mentioned there.

*Page 3: Insert a table of the earthquakes described.*
A table was added to the manuscript with the requested data.

*Page 4: Insert some stratigraphic columns and shear wave velocity profiles.*
We do not think it adequate to include stratigraphic columns in our manuscript. We believe that a discussion of the different soil types present in Armenia and of the subsoil structure would deviate our manuscript from its main subject: the vulnerability of the building stock in the city. However, we added a sentence to indicate that a thorough discussion of that subject was included in part I of our study. In contrast, we did prepare a table with data of the soil profiles at two sites in the city of Armenia and included it in the revised manuscript. Those two sites are representative of the range of variations of shear wave velocity and soil thickness throughout the city. This addition required the modification of Figure 2, where the location of those two sites is now indicated. We took this opportunity to improve additional details in Figure 2.

*Figure 2: Enter the text in English.*
We do apologize for this oversight. We do not understand how could the caption of Figure 2 manage to slip into the submitted document in Spanish. This blunder has been corrected.

We thank again Anonymous Referee #1 for his/her remarks that were helpful to improve and correct our manuscript. Thanks to him/her our revised manuscript has been improved relative to our original submission.

**Comments by Dr. Francesco Panzera, Referee #2**

First, we would like to thank Dr. Panzera for his review of our manuscript. His report includes a general evaluation of our manuscript, four comments pointing to details that could be improved in our text, and a general comment concerning a more significant topic. In the following, we recall each of the reviewer's objections and describe the way in which our revised manuscript takes into account the reviewer's comments. The changes made to the manuscript to solve the concerns of Dr. Panzera have been highlighted in green in the marked-up version of our revised manuscript.

*General evaluation of the manuscript.*
Dr. Panzera first describes briefly the contents and intent of our manuscript. We thank him for his positive judgment of our work.

*Line 57: add "of buildings"…*
Modified as requested.

*Line 82: add some worldwide references…*
The referee suggests adding to our reference list four papers that cover similar subjects as our manuscript and highlight cases that occurred in areas removed from our case of study. We thank Dr. Panzera for providing the links to the publications, which eased access to those papers. After reading them, we added references to three of them. The fourth paper deals with a subject that is removed from the substance of our manuscript. Its emphasis is on equations to predict intensities and the Swiss geologic-tectonic classes and is not a useful reference for our case. However, we have added a fourth reference to another study that compares ground motion amplification with observed damage estimates for a case study in Greece. The addition of these references has broadened significantly the geography of referenced papers in our manuscript. In addition, they simplify the way (through the references included in those four papers) for interested readers to access more papers on our subject.

*Line 86: change "relatively small"…*
Modified as requested.

*Line 184: "the geological formations…", please specify the used geologic map scale.*
The referee is correct. It is important to indicate the scale of the geological map. The revised manuscript corrects this oversight.

*General comment.*
Finally, in a general comment, Dr. Panzera suggests us to improve the discussion around the possibility of site effects contributing to irregular damage distribution. We thank him for this remark that identifies a point where our manuscript could improve. In response to this comment, we point out here that this manuscript is the second part of a study on ground motion of the 1999, Armenia, earthquake and the understanding of the factors which played a role in the damage observed during that event. A detailed account of ground motion, subsoil structure, and site effects in Armenia is the main subject of part I of our study. However, the general comment from Dr. Panzera indicates that our manuscript could better emphasize that site effects were dealt with in part I of our study. In addition, our current manuscript, part II, does require to be self-contained. For this reason, we have added some comments regarding the possible double resonance effect (soft soil resonance coupled to building resonance) and the way in which we consider that site effects contributed to the observed damage.

Finally, we thank again Dr. Panzera for his remarks and the careful evaluation of our manuscript. His review identified weak points in our manuscript. As a result of his review, our paper has been improved.